# The Anti-Constipation Effect of Bifidobacterium Longum W11 Is Likely Due to a Key Genetic Factor Governing Arabinan Utilization

**DOI:** 10.3390/microorganisms12081626

**Published:** 2024-08-09

**Authors:** Francesco Di Pierro, Nicola Zerbinati, Massimiliano Cazzaniga, Alexander Bertuccioli, Chiara Maria Palazzi, Ilaria Cavecchia, Mariarosaria Matera, Edoardo Labrini, Valeria Sagheddu, Sara Soldi

**Affiliations:** 1Microbiota International Clinical Society, 10123 Torino, Italy; alexander.bertuccioli@uniurb.it (A.B.);; 2Scientific & Research Department, Velleja Research, 20125 Milano, Italy; 3Department of Medicine and Technological Innovation, University of Insubria, 21100 Varese, Italy; 4Department of Biomolecular Sciences, University of Urbino Carlo Bo, 61122 Urbino, Italy; 5Microbiomic Department, Koelliker Hospital, 10134 Turin, Italy; 6Department of Paediatric Emergencies, Misericordia Hospital, 58100 Grosseto, Italy; 7AAT—Advanced Analytical Technologies, Fiorenzuola d’Arda, 29017 Piacenza, Italy; edoardo.labrini@aat-taa.eu (E.L.); sara.soldi@aat-taa.eu (S.S.)

**Keywords:** *abfA*, *abfB*, arabinofuranoside, gastrointestinal motility

## Abstract

Recent investigations have highlighted, both experimentally and clinically, that probiotic strains equipped with arabinofuranosidase, in particular *abfA* and *abfB*, favor regular intestinal motility, thus counteracting constipation. By analyzing the gene expression and the proliferative response in the presence of arabinan of the probiotic *B. longum* W11, a strain previously validated as an anti-constipation probiotic, we have speculated that its response mechanism to arabinan can effectively explain its clinical action. Our approach could be used in the future to select probiotics endowed with arabinofuranosidase-related anti-constipation effects.

## 1. Introduction

Orally administered probiotics are widely used to alleviate functional constipation, a disorder affecting about 14% of the adult population and 9.5% of children, and recent meta-analysis reported that probiotic use moderately increases stool frequency and reduces the whole-gut transit time [1,2,3,4]. Probiotic ingestion can modify the gut microbial fermentation leading to a short-chain fatty acids (SCFAs) profile that interacts differently with the host immune system and enteric nervous system, thus ameliorating gut motility [5]. However, the effect of probiotics in functional constipation seems to be strain dependent [6]. Recently, Zhang et al. showed that *B. longum* strains possessing the encoding arabinofuranosidases cluster *abfA* can ameliorate functional constipation through enhanced arabinan utilization in the gut in both mice and humans [7]. Their results would seem to suggest the existence of a potentially specific therapeutic tool (i.e., *B. longum* equipped with *abfA*, administered together with arabinan fiber) for functional constipation. The *B. longum* strain W11 (LMG P-21586), a probiotic presenting intrinsic and non-transferable resistance to rifaximin [8,9,10], has also been described as being effective in treating functional constipation [11,12,13]. We have, therefore, investigated whether the results describing the role of the strain W11 in constipation could be linked to the presence and functionality of the same gene cluster. Three genes putatively encoding arabinofuranosidases have been described (*abfI*, *abfA*, and *abfB*) [14]. However, we have focused only on *abfA* and *abfB* as *abfI* has been shown not to affect arabinofuranosyl residues. As shown by our methods and results, we have evaluated both the presence of two arabinofuranosidases clusters (*abfA* and *abfB*) and their action in metabolizing arabinan which also supports the in vitro growth of the probiotic strain. The detection of the genes in the strain genome, their overexpression in arabinan-enriched medium, and the strain condition-specific in vitro growth have provided a probable explanation for its effectiveness in constipation.

## 2. Materials and Methods

### 2.1. In Silico Evaluation of abfA and abfB Genes Presence in B. Longum W11 Genome

For the in silico evaluation of the presence of the genes *abfA* and, *abfB*, we conducted a search using BV-BRC ver 3.35.5 software employing the deposited sequence of the *B. longum* strain W11 (NCBI Accession n° PRJNA356203).

### 2.2. Viable Counts of B. Longum W11 in Different Sugar-Restricted Media

Growth assays of *B. longum* W11 were performed in sugar-restricted basal medium [15] alone (1.0% Bacto peptone, 0.5% Bacto yeast extract, 0.5% sodium acetate trihydrate, 0.2% diammonium hydrogen citrate, 0.08% L-cysteine hydrochloride monohydrate, 0.02% magnesium sulfate heptahydrate, 1.36% L-ascorbic acid, 0.44% sodium carbonate anhydrous; all reagents from Merck, Milan, Italy) or supplemented with 2.0% glucose or arabinan (from sugar beet; Megazyme, MI, USA) [16]. The first viable counts were performed at T0 and after 48 h of incubation at 37 °C under anaerobic conditions. We performed viable counts according to ISO 29981/IDF 220:2010, a method for the selective enumeration of presumptive bifidobacteria in milk products by using a colony count technique at 37 °C under anaerobic conditions [17].

### 2.3. RNA Extraction, Quantification and Retro Transcription for Downstream Analysis

An appropriate volume of the T0 and of the three tested conditions (no sugar, glucose and arabinan) at T16, T24 and T48, was collected for the total RNA extraction. The volume was chosen to reach the cell concentration of 1–2 × 109 bacteria. The cultures were centrifuged, supernatants discarded, and the pellets underwent a lysis phase with 10 mg/mL of lysozyme and 300 U/mL of mutanolysin for 1 h and 30 min at 37 °C (enzymes from Merck, Milan, Italy). The Aurum™ Total RNA Mini Kit (Bio-Rad, Hercules, CA, USA) was used for the isolation and purification of total RNA. The extracted RNA was quantified through Nanodrop and the Qubit RNA HS assay (Invitrogen, Segrate, Italy). DNA contamination was checked with the Qubit ds DNA HS assay (Invitrogen, Segrate, Italy). A treatment with DNase was performed to remove gDNA traces. RNA underwent the retro transcription using the iScript™ cDNA Synthesis Kit (Bio-Rad, Hercules, CA, USA) following manufacturer instructions.

### 2.4. qPCR Amplification 

Quantitative PCR (qPCR) analysis was performed using StepOne Plus Instrument (Applied Biosystems, Waltham, MA, USA) with fluorescence signal detection (SYBR green) after each amplification cycle. PCRs were performed in a 25-μL reaction mixture: 12.5 μL of Bio-Rad SsoAdvanced Universal mix (Bio-Rad, Hercules, CA, USA), 0.1 (for 16S DNA) or 1 (target genes) ng of cDNA, each forward and reverse primer at the proper concentration and sufficient nuclease-free water to obtain a final volume of 25 μL. Negative controls for each primer set were included in each run. Primer sets (16S DNA housekeeping gene, *abfA* and *abfB*) and thermal protocols were assessed as previously described [14]. Gene expression data were analyzed through the ΔΔCt method by comparing the relative concentration of the target gene in the treated sample to that of the control one (*B. longum* W11 at T0). Tests were performed in triplicate.

### 2.5. Statistical Analysis

We performed statistical analysis using the Brown–Forsythe method (Prism- GraphPad, v8). Statistical significance was set for *p* value < 0.05.

## 3. Results

We performed an in silico analysis of the deposited genome sequence of *B. longum* W11 to find the *abfA* and *abfB* genes. As shown in Figure 1, the bioinformatic research highlighted the presence of the two genes. We then evaluated the *B. longum* W11 viable counts in different sugar-restricted media containing glucose or arabinan at 2% (*w*/*v*). As shown in Table 1, the total counts obtained for *B. longum* W11 in arabinan-supplemented medium, in comparison with others, demonstrate that the strain actively metabolized this sugar, producing a proliferative effect of more than 1 log. We then quantified the *abfA* and *abfB* expression in the presence of arabinan at different time points. The precise RNA and DNA quantifications are available in Appendix A. As shown in Figure 2, after 16 h of incubation, the two genes *abfA* and *abfB* were overexpressed in the sugar-restricted medium supplemented with 2% of arabinan. After 24 h (as after 48 h), the overexpression detected at 16 h was no longer observable. Indeed, as previously described [7], *abfA* and *abfB* seem to be expressed more in the first phase of bacterial growth in the presence of arabinan.

## 4. Discussion

As recently described, *B. longum* strains holding the *abfA* cluster can ameliorate functional constipation in animals and in humans through enhanced arabinan utilization, metabolically yielding acetate, butyrate, chenodeoxycholic acid, and uracil [7]. Acetate and butyrate are two SCFAs implicated in improving intestinal motility either by interacting with G-protein-coupled receptor 41 (GPR41) and G-protein-coupled receptor 43 (GPR43) or directly acting on colonic smooth muscle [18,19]. Acetate and butyrate producers are associated with the bile acid level (i.e., chenodeoxycholic acid) in the gut [20,21]. For patients with irritable bowel syndrome constipation-type (IBS-C), it is reported that treatment with chenodeoxycholic acid can increase stool frequency, improve stool consistency, and improve ease of stool passage by acting on the membrane-bound G protein-coupled bile-acid receptor (e.g., TGR5) on enterocytes [22,23,24]. Similarly, a has been established between increased microbiota-induced uracil levels and decreased IBS-C disease activity [20]. One can assume that, in the proximal part of the colon, the glucose supply is sufficient to guarantee the microbial production of SCFAs, bile acids, and uracil. Differently, in the distal colon, the glucose supply should be exhausted. In this circumstance, microorganisms equipped with *abfA* clusters could stably proliferate and produce the beneficial metabolites above described.

*B. longum* W11, previously reported to be clinically effective against constipation [10,11,12,13] possesses the *abfA* and *abfB* genes and increases the expression of both *abfA* and *abfB* genes in the presence of arabinan. Moreover, *B. longum* W11 proliferates in the presence of arabinan. The results seem to be quite specific since the same strain (*B. longum* W11) does not increase the amounts of these two genes and does not proliferate in the presence of glucose. Our results should seem then to indicate a possible mechanism of action by which the strain W11 counteracts constipation. Moreover, our study demonstrates that *B. longum* W11 can be considered a “precision probiotics” [25] since, on a genetic basis, it presents different and concomitant peculiarities (rifaximin resistant and anti-constipation). Of course, our results do not exclude that other possible mechanisms of action could take part in ameliorating such clinical outcome. Having established a certain relationship between the presence of arabinofuranosidases and the anti-constipation effect, however, does not mean that any bacterial strain, with reference to those with evident probiotic characteristics, equipped with the same genes, is capable of anti-constipation efficacy. For the effect to be clinically observed, it is necessary for the genes to be functional, that the bacterium equipped with them remains viable inside the host for a sufficiently long time, and that it finds, in the ecosystem in which it has been introduced, an appropriate quantity of specific fiber, arabinan, or its potential precursors such as pectin or arabinoxylans [26,27]. As amply demonstrated, the *abfA* and *abfB* genes are not present only in the species *B. longum* [7]. By in silico analysis, we have explored the presence of this gene cluster in the probiotic genera like *Bacillus* and *Bifidobacterium* and in what until recently was simply called *Lactobacillus* [28]. Our analyses have revealed that this gene cluster is quite present among probiotics, especially among bifidobacteria. Among the probably most studied and worldwide marketed probiotic strains, such as the *Bacillus clausii* ENT-pro^®^strains [29], *B. animalis lactis* BB-12^®^ [30], and *L. rhamnosus* GG^®^ [31], the latter, reviewed as ineffective in functional constipation [32,33], does not present arabinofuranosidase genes. In contrast, *B. clausii* strains and BB-12^®^ have both been described to have clinical anti-constipation effects [34,35,36,37,38].

## 5. Conclusions

Although our work was limited mainly to the *B. longum* W11 strain, it lays the groundwork for identifying a rationale between the strain’s ability to metabolize arabinan and the consequent overexpression of the *abfA* and *abfB* genes. This finding could reveal the mechanism underlying the anti-constipation action exerted by the strain under investigation. As such, this approach might be used to select or better characterize other probiotics with the same function. Last but not least, to our knowledge, our work identifies for the first time the possibility for a “precision probiotic” to be characterized by two completely unrelated features.

## Figures and Tables

**Figure 1 microorganisms-12-01626-f001:**
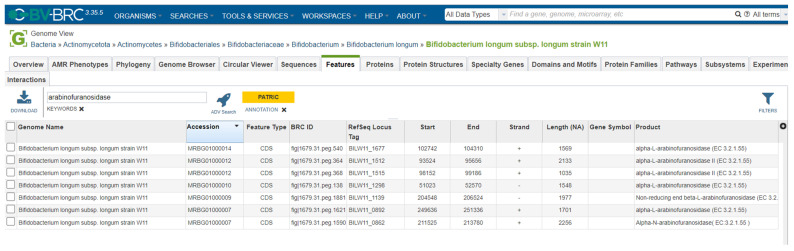
In silico evaluation of the presence of *abfA* and *abfB* genes.

**Figure 2 microorganisms-12-01626-f002:**
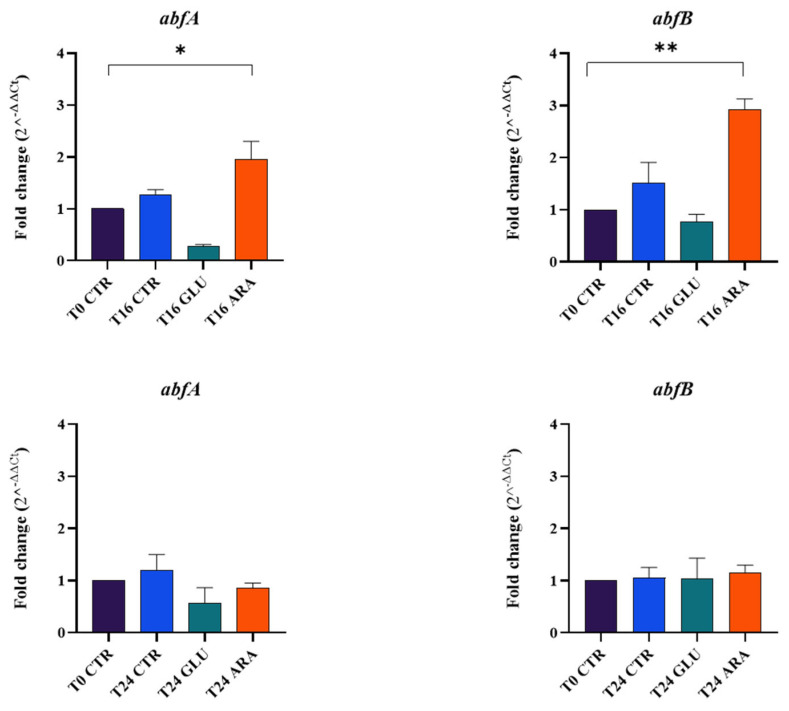
*abfA* and *abfB* genes’ expression at T0, T16 and T24 in the tested conditions. * *p* < 0.05; ** *p* < 0.01.

**Table 1 microorganisms-12-01626-t001:** Viable counts at two timepoints (T0 and T24).

**Sugar restricted, T0**					**Sugar restricted, T24**				
	−5	−6	CFU/mL	Log (CFUs)	−5	−6	CFU/mL	Log (CFUs)	DL (CFUs)
BLW11	187	15	1.8 × 10^7^	7.3	161	23	1.67 × 10^7^	7.2	0.0
**Sugar restricted + glucose, T0**					**Sugar restricted + glucose, T24**				
	−5	−6	CFU/mL	Log (CFUs)	−7	−8	CFU/mL	Log (CFUs)	DL (CFUs)
BLW11	175	14	1.7 × 10^7^	7.2	44	6	4.55 × 10^7^	7.7	0.4
**Sugar restricted + arabinan, T0**					**Sugar restricted + arabinan, T24**				
	−5	−6	CFU/mL	Log (CFUs)	−7	−8	CFU/mL	Log (CFUs)	DL (CFUs)
BLW11	179	18	1.8 × 10^7^	7.3	82	11	8.45 × 10^7^	8.9	1.7

BLW11: *B. longum* W11; DL: Delta Log; CFU: Colony Forming Units.

## Data Availability

The raw data supporting the conclusions of this article will be made available by the authors on request.

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
