# Peer review of "The Anti-Constipation Effect of Bifidobacterium Longum W11 Is Likely Due to a Key Genetic Factor Governing Arabinan Utilization"

_microorganisms, 2024, doi:10.3390/microorganisms12081626_

Round 1

Reviewer 1 Report

Comments and Suggestions for Authors

This is a very specialized research, aiming  to document the  existence of two of the three genes encoding arabinofuranosidases, the abfA and abfB in Bifidobacterium longum W11, known to ameliorate functional constipation.

These results open the way for similar search in other strains as to whether they are ably to metabolize arabinan through the expression of these genes and thus positive implicated in the alleviation of constipation.

Minor comment

although there are references in irritable bower syndrome, I would suggest authors to remain to simply "functional constipation" [para 1, line 15], since in IBD other psychosomatic mechanisms theoretically are implicated, too

Reviewer 2 Report

Comments and Suggestions for Authors

The manuscript by Francesco Di Pierro et al. is largely a repeated work of "A key genetic factor governing arabinan utilization in the gut microbiome alleviates constipation" published in Cell Host Microbe. I have not seen any new ideas and new findings in it. I suggest to reject it. 
